# Properties of Calcium Sulfoaluminate Cement Mortar Modified by Hydroxyethyl Methyl Celluloses with Different Degrees of Substitution

**DOI:** 10.3390/molecules26082136

**Published:** 2021-04-08

**Authors:** Shaokang Zhang, Ru Wang, Linglin Xu, Andreas Hecker, Horst-Michael Ludwig, Peiming Wang

**Affiliations:** 1Key Laboratory of Advanced Civil Engineering Materials (Tongji University), Ministry of Education, School of Materials Science and Engineering, Tongji University, Shanghai 201804, China; 1810061@tongji.edu.cn (S.Z.); xulinglin@126.com (L.X.); tjwpm@126.com (P.W.); 2F. A. Finger for Building Materials, Bauhaus-University Weimar, D-99421 Weimar, Germany; andreas.hecker@uni-weimar.de (A.H.); horst-michael.ludwig@uni-weimar.de (H.-M.L.)

**Keywords:** calcium sulfoaluminate cement, hydroxyethyl methyl cellulose, degree of substitution, mechanical properties, porosity

## Abstract

This paper studies the influence of hydroxyethyl methyl cellulose (HEMC) on the properties of calcium sulfoaluminate (CSA) cement mortar. In order to explore the applicability of different HEMCs in CSA cement mortars, HEMCs with higher and lower molar substitution (MS)/degree of substitution (DS) and polyacrylamide (PAAm) modification were used. At the same time, two kinds of CSA cements with different contents of ye’elimite were selected. Properties of cement mortar in fresh and hardened states were investigated, including the fluidity, consistency and water-retention rate of fresh mortar and the compressive strength, flexural strength, tensile bond strength and dry shrinkage rate of hardened mortar. The porosity and pore size distribution were also analyzed by mercury intrusion porosimetry (MIP). Results show that HEMCs improve the fresh state properties and tensile bond strength of both types of CSA cement mortars. However, the compressive strength of CSA cement mortars is greatly decreased by the addition of HEMCs, and the flexural strength is decreased slightly. The MIP measurement shows that HEMCs increase the amount of micron-level pores and the porosity. The HEMCs with different MS/DS have different effects on the improvement of tensile bond strength in different CSA cement mortars. PAAm modification can improve the tensile bond strength of HEMC-modified CSA cement mortar.

## 1. Introduction

The cement industry is responsible for 5% of global anthropogenic greenhouse gas emissions [1,2]. Meanwhile, it also consumes 10–11 EJ of energy resources annually, which accounts for 2–3% of the use of global primary energy [3]. In order to solve this problem, it is necessary to reduce the lime content in raw material [4] and to lower the calcination temperature [5] of clinker. CSA cement has the advantages of low-CO_2_ emissions and low energy consumption during production, which exactly meets the needs of energy conservation and emissions reduction.

With the rapid development of commercial mortar, CSA cement has been introduced to commercial mortar and is widely used in job sites due to its excellent properties, such as high early strength, rapid hardening and so on. At the same time, CSA cement can be used as cementitious material in self-leveling mortar, concrete grouting material, ceramic tile adhesive, rapid repair material, etc. In those areas, cellulose ether is usually regarded as a good water-retention and thickening agent, and it is widely applied to improve water retention and cohesion. Hydroxyethyl methyl cellulose (HEMC) is a kind of commonly used cellulose ether (CE) in construction. It has been confirmed that HEMC has effects on the workability [6], rheological properties [7], mechanical properties [8], hydration [9] and pore structure [10] of Portland cement mortar. The mechanical properties [11], early hydration [12] and the existence state of water [13] in CSA cement mortar can be also influenced by methylcellulose (MC) and HEMC. The type of ether, degree of substitution (DS), molar substitution (MS) and average molecular weight (M_w_) of CE molecules are always regarded as the characterization of the chemistry of CEs [14]. Ou et al. [9,15] reported that the molecular parameters of CEs (M_w_, type and viscosity) can affect the early-age hydration kinetics and mechanical properties of Portland cement mortar. Patural et al. [16] proposed that the physico-chemical parameters of CE, like M_w_, DS, etc., have a strong influence on the water retention capacity of Portland cement mortar.

The DS of CE is a very important molecular parameter of CE structure; it can be defined as the average number of hydroxyl groups of an anhydroglucose unit occupied by ether groups. The maximal value of DS is three. The MS of CE means the average number of ether groups (for CEs that contain only one type of ether), or of each type of ether group (for CEs that contain more than one type of ether group), per anhydroglucose unit, and it is also an important molecular parameter. Compared to the value of DS, the value of MS has no theoretical upper limit. Weyer et al. [17] proposed that the lower the DS, the stronger the retardation of C_2_S/C_3_S hydration. Pourchez et al. [18,19] investigated the influence of hydroxyethyl cellulose (HEC) on the hydration of C_3_S and C_3_A and found that the substitution groups and DS of HEC seem to be more important controlling factors on C_3_S and C_3_A hydration than molecular mass. Kim et al. [20] also proposed that the increasing MS/DS of HPMC causes the increase in hydrophilicity and the lowering of hydrogen bonding. But no literature has confirmed that CE with different MS/DS would have influence on the properties of cement mortar, even for Portland cement mortar.

The effect of CE on the pore size and porosity of Portland-cement-based materials was researched; it is a key role that affects mechanical properties. Ou et al. [21] compared the effects of four CEs on the pore structure of ordinary Portland cement paste and found that the total porosity of cement pastes containing the four CEs were significantly higher than that of pure cement paste and that the total porosity of cement pastes containing HEC or low viscosity CEs were low. By changing the surface tension and viscosity of the liquid phase and strengthening the liquid film between air voids in cement pastes, CEs affect the formation, diameter evolution and upward movement of air voids and the pore structure of hardening cement paste. CEs affect the pore structure of cement paste mainly through their effects on the evolvement of small air voids into bigger ones when the pore diameter is below 70 nm and their effects on the entrainment and stabilization of air voids when the pore diameter is above 70 nm. Pourchez et al. [22,23] studied the effect of HPMC and HEC on the pore structure of ordinary Portland cement pastes and found that CE chemistry has a strong effect on pore structure development, which can be summarized as a high porosity for cement pastes containing HPMCs, a medium porosity for cement pastes containing high-molecular-mass HECs and a lower porosity for cement pastes containing low-molecular-mass HECs and pure cement pastes.

The differences in mineral composition between CSA and Portland cement can influence the behavior of CE in cement mortar [24]. Considering the impact that would be brought by the difference, it is necessary to investigate the influence of CE on the properties of CSA cement mortar. Meanwhile, few studies have researched the influence of CE on the properties of CSA cement mortar, especially for CEs with different chemistry structures or with polyacrylamide (PAAm) modification. Therefore, in this investigation, HEMCs with higher and lower MS/DS and PAAm-modified HEMC were chosen as admixtures. Two types of CSA cements with different content of ye’elimite were chosen as cementitious materials. A series of properties, including fresh state properties and hardened state properties of the cement mortar, were tested. The porosity and pore size distribution of cement mortars were also analyzed by mercury intrusion porosimetry (MIP) measurement.

## 2. Materials and Methods

### 2.1. Raw Cement

Two types of CSA (CSA1 and CSA2) cements were used, both of which have different mineral compositions; both are widely used in China. The chemical compositions of the CSA cements, determined by X-ray fluorescence (XRF), are listed in Table 1. The mineral compositions of the CSA cements were analyzed by QXRD analysis through Profex/BGMN software package (Ver. 3.10) [25] and are shown in Table 2. The particle size distribution of CSA cements is shown in Figure 1.

### 2.2. Cellulose Ether

Three kinds of HEMCs, labeled HEMC1, HEMC2 and HEMC3, were used. The physico-chemical properties of HEMC1 and HEMC2 are listed in Table 3. HEMC3 was obtained by adding 5% polyacrylamide (PAAm) into HEMC2. Quartz sand with a diameter of 0.1–0.5 mm and tap water were used.

### 2.3. Proportion and Mixing

The mortar specimens were prepared with a HEMC/cement mass ratio (*m*_HEMC_/*m*_c_) of 0.3%, a cement/sand mass ratio of 3:7, which was determined based on the mixture ratio of common ceramic tile adhesive, and a water/cement mass ratio of 0.60, which was determined according to experiments that cause the mortar to exhibit good workability. The mixing procedure was as following: the dry powders were blended for 30 s at low speed, then water was added and mixed for 30 s at low speed and, in the next 30 s slow-speed mixing segment, the sand was added gradually, followed by 30 s of mixing at high speed, a break of 90 s and, subsequently, an extra 60 s mixing at high speed.

### 2.4. Water Retention

The retention of water in fresh mortar was described by the percentage of water that remained in the mortar after a brief draining of water through filter paper [26]. The water-retention rate of fresh mortar was tested according to the Chinese standard JGJ 70-2009.

### 2.5. Consistency

The consistency of the fresh mortar was measured by using a mortar consistency meter according to the Chinese standard JGJ 70-2009. A conical container was filled with the mortar to a level that was 1 cm below its rim. The mortar was compacted inside the container using the tamping rod. The container was placed over the base and below the penetrating cone of the apparatus. The penetrating cone was lowered so that the apex of the penetrating cone just touched the mortar surface. The penetrating cone was clamped at this position. Then the cone was released and allowed to sink into the mortar mix. The final penetration depth of the cone in the mortar was recorded as the consistency.

### 2.6. Fluidity

The fluidity of fresh mortar was measured according to GB/T 2419–2005. A cone-shaped metal ring was filled with fresh cement mortar on a shock table, and after lifting the ring, the fresh mortar was subjected to 25 drops of the table within 25 s. The final diameter of the fresh mortar was the so-called flow table value, which was used to represent the fluidity of the fresh mortar.

### 2.7. Compressive and Flexural Strengths

The flexural and compressive strength tests of cement mortar were conducted according to GB/T 17671–1999. The specimen size was 40 mm × 40 mm × 160 mm. The specimens were demolded after 1 day of curing at (20 ± 2) °C/90% RH and then cured at (20 ± 5) °C/(60 ± 5)% RH to targeted ages. The compressive strength was tested at a loading rate of 2.4 kN/s at the age of 1, 3 and 28 days by using a material testing machine, while the flexural strength was determined by the 3-point bending test at a loading speed of 50 N/s at 1, 3 and 28 days.

### 2.8. Tensile Bond Strength

The specimens for the tensile bond strength measurement were prepared according to JC/T 985-2005. The fresh mixtures were cast on a concrete board that complied with JC/T 547-2005. Eight specimens were prepared for each mixture. The size of the sample was 50 mm × 50 mm × 5 mm. The specimens were cured under (20 ± 2) °C/60% RH for 24 h, then unmolded and cured in the same environment sequentially. The tensile bond strength was tested at 3, 7, and 28 days, respectively. One day before the test, the specimens were glued to an iron block by epoxy resin. The measurement of tensile bond strength was determined according to JC/T 985-2005.

### 2.9. Dry Shrinkage Rate

The dry shrinkage rate was tested in accordance with JGJ/T 70-2009. The initial length was measured after 24 h of curing. Subsequently, the length of each specimen was recorded after further curing for 1, 3, 7, 14 and 28 days. The average dry shrinkage rate of three samples was taken as the representative value for every formulation.

### 2.10. MIP Analysis

A mercury intrusion porosimeter was used to measure the porosity characteristics of CSA cement mortars. The pore diameter was calculated by using the Washburn equation [27]. The curing age of samples used in MIP tests was 28 days. Before test, the samples had to be hand-formed into pellets with a size of about 5–7 mm, and the hydration of the pellets was terminated by ethyl alcohol for a week. During the process of terminating the hydration, the ethyl alcohol was changed every two days. After the process of terminating the hydration ended, the samples were dried in a 40 °C oven for 48 h. The measurement was carried out on Quantachrome Poremaster GT-60 (Quantachrome, Boynton beach, FL, USA), and the osmotic pressure was set at the range of 29–15,242 psi. In this experiment, the pore diameters were tested in an accurate range of 14 nm–7.2 µm.

## 3. Results and Discussion

### 3.1. Fresh State Properties

The water-retention rate, consistency and fluidity of fresh CSA cement mortars are listed in Table 4. At the same experimental condition, the addition of HEMCs leads to an increase in the water-retention rate, consistency and fluidity of fresh CSA cement mortars. It seems that HEMCs with higher MS/DS, lower MS/DS and PAAm modification have the same impact on water-retention capacity. The HEMCs improve water retention up to >99.4%. This means that HEMCs also exhibit excellent water retention when it is used in CSA cement mortars. The viscosity and M_w_ of all the HEMCs used is very close, so the results confirm that the water-retention capacity of HEMCs does not highly depend on the MS/DS.

The addition of HEMCs leads to an increase in the consistency of fresh cement mortars. According to reference [28], when the concentration of HEMC is between 0.20% and 0.29%, an increase in the concentration of HEMC leads to an increase in the consistency of fresh Portland cement mortar. It is clear that the HEMCs have a similar function in CSA cement mortar.

Fluidity is a very important parameter to characterize the workability of fresh cement mortar, which is linked with its consistency [29,30]. Due to the rapid hardening property, the fluidity values of fresh CSA cement mortars are very low, whereas the addition of HEMCs can increase the fluidity of fresh CSA cement mortars and avoid the bleeding, i.e., improve the workability of the fresh mortars significantly.

### 3.2. Hardened State Properties

#### 3.2.1. Compressive Strength

The compressive strength of CSA cement mortars with and without HEMCs is illustrated in Figure 2. It can be observed that the compressive strength of plain CSA1 cement mortar is much higher than that of plain CSA2 cement mortar, especially at early ages. The early strength of CSA cement mortar can be attributed to bonding and interlocking between ettringite (AFt) crystals [31,32]. The amorphous phases like AH_3_ and C-S-H gel can contribute to later-age strength due to the formation of a dense and low-porosity matrix by filling in the crystal skeleton [33]. Meanwhile, ye’elimite can rapidly react with calcium sulfate to form ettringite [34]. According to the information in Table 2, the content of ye’elimite of CSA1 (28%) is much higher than that of CSA2 (16%). That is probably the main reason that leads to the compressive strength difference.

After adding HEMC, the compressive strength of both types of CSA cement mortars decreases. The strength reduction of both types of CSA cement mortars at different curing ages is around 50%, which is in agreement with the result for Portland cement mortar [29]. HEMCs can entrain air bubbles into fresh cement mortar during the process of stirring, increasing the porosity of hardened cement mortar, and, thus, the compressive strength decreases, which will be discussed in the following porosity discussion section.

In addition, the compressive strength of HEMC1-modified CSA1 cement mortars is higher than that of HEMC2-modified CSA1 cement mortars, indicating that HEMCs with higher MS/DS decreases the compressive strength more significantly at all curing ages. At the curing ages of 1 and 3 days, the compressive strength of HEMC2-modified CSA2 cement mortars is higher than that of HEMC1-modified CSA2 cement mortars. This means that HEMC with lower MS/DS decreases the compressive strength of CSA2 cement mortars more significantly, at the early ages (1 and 3 days). At the later age (28 days), HEMC1-modified CSA2 cement mortars exhibit higher compressive strength than HEMC1-modified CSA2 cement mortar. This means that HEMC with higher MS/DS decreases the compressive strength of CSA2 cement mortars more significantly at the later age (28 days). In all, HEMCs with different MS/DS do have different impacts on the properties of CSA cement mortar, and the impacts are related to the type of CSA cement. The influence of HEMCs with different MS/DS on the properties of CSA2 cement mortar is also related to the curing age. At all curing ages, the compressive strength of HEMC2-modified CSA cement mortars is higher than that of HEMC3-modified CSA cement mortars. According to the research [35], PAAm can improve the compressive strength of Portland cement mortar at a very low dosage (0.05%). But in this investigation, PAAm can not change the effect of HEMCs on the compressive strength of CSA cement mortars and even decreases the compressive strength to some extent.

#### 3.2.2. Flexural Strength

The flexural strength of CSA cement mortars with and without HEMCs is illustrated in Figure 3. It can be seen that CSA cement mortar made from CSA1 cement that has higher ye’elimite content exhibited higher flexural strength at all curing ages. It can also be seen that HEMCs slightly decrease the flexural strength of CSA cement mortar. At all curing ages, the flexural strength of HEMC1-modified CSA1 cement mortar is higher than that of HEMC2-modified CSA1 cement mortar, indicating that HEMC with higher MS/DS decreases the flexural strength more significantly. In contrast, the flexural strength of HEMC1-modified CSA2 cement mortar is lower than that of HEMC2-modified CSA2 cement mortar at all curing ages. It means that HEMC with lower MS/DS decreases the flexural strength more significantly at all curing ages.

The flexural strength of HEMC3-modified CSA1 cement mortar is the lowest, meaning that PAAm can enhance the negative effect of HEMC2 on the flexural strength of CSA1 cement mortar. In contrast, HEMC3-modified CSA2 cement mortar is slightly higher than that of HEMC2-modified CSA2 cement mortar. It indicates that PAAm weakens the effect of HEMC2 on the flexural strength of CSA2 cement mortar. Due to the increase in porosity, the flexural strength of CSA cement mortars decreases. The difference in sample failure mode leads to a reduced degree of flexural strength compared to compressive strength [36].

#### 3.2.3. Ratio of Compressive Strength to Flexural Strength

In order to evaluate the flexibility of different CSA cement mortars with and without HEMCs, the ratio of compressive strength to flexural strength was calculated. The lower the ratio, the better the flexibility. The result is shown in Figure 4. It can be seen that, compared to CSA1 cement mortar, CSA2 cement mortar exhibited better flexibility at 1 and 3 days. HEMCs can improve the flexibility of different CSA cement mortars. There is no significant difference among the three kinds of HEMC-modified CSA1 cement mortars in the ratio of compressive strength to flexural strength. It means that the MS/DS of HEMC and PAAm modification have a minor effect on the flexibility of HEMC-modified CSA1 cement mortar. But the ratio of compressive strength to flexural strength in HEMC3-modified CSA2 cement mortar is the lowest among the three kinds of modified CSA2 cement mortars. It means that HEMC modified by PAAm can improve the flexibility of CSA2 cement mortar significantly. At early ages (1 and 3 days), the ratio of compressive strength to flexural strength in HEMC2-modified CSA2 cement mortar was similar to that in HEMC1-modified CSA2 cement mortar, meaning MS/DS had no significant effect on the flexibility of CSA2 cement mortar at early ages. At the age of 28 days, the ratio of compressive strength to flexural strength in HEMC2-modified CSA2 cement mortar was lower than that of HEMC1-modified CSA2 cement mortar. That means that HEMC with higher MS/DS can improve the flexibility of CSA2 cement mortar at later ages. At all curing ages, the ratio of compressive strength to flexural strength in HEMC3-modified CSA2 cement mortar was lower than that in HEMC2-modified CSA2 cement mortar. This means that the addition of PAAm enhances the positive effect of HEMC2 on the improvement of flexibility of CSA2 cement mortar.

#### 3.2.4. Tensile Bond Strength

The tensile bond strength of CSA cement mortars with and without HEMCs is illustrated in Figure 5. It is clear that the tensile bond strength of CSA1 cement mortar is higher than that of CSA2. HEMCs can increase the tensile bond strength of CSA cement mortars, but the effect of different HEMCs in CSA1 and CSA2 cement mortar varies. For example, the tensile bond strength of HEMC1-modified CSA1 cement mortar was higher than that of HEMC2-modified CSA1 cement mortar. This means that the lower MS/DS of HEMC contributes to the improvement of tensile bond strength. Meanwhile, the tensile bond strength of HEMC3-modified CSA1 cement mortar was higher than that of HEMC2-modified CSA1 cement mortar, indicating that PAAm favors the improvement of HEMC2 on the tensile bond strength of CSA1 cement mortar.

As for the tensile bond strength of CSA2 cement mortar, the effect of HEMC with different MS/DS on the tensile bond strength has changed, but the effect of PAAm has not changed. For example, the tensile bond strength of HEMC2-modified CSA2 cement mortar is higher than that of HEMC1-modified ones. It shows that the higher MS/DS of HEMC favors the improvement in the tensile bond strength of CSA2 cement. Meanwhile, the tensile bond strength of HEMC3-modified cement mortar is still higher than that of HEMC2-modified cement mortar when CSA2 cement is used as cementitious material. It means that PAAm also has a positive effect on the improvement of tensile bond strength of HEMC2-modified CSA2 cement mortar.

According to the references, the mechanism of the adhesion of cement mortar can be concluded to be mechanical interlocking between the adhesive and the concrete substrate [28]. On the one hand, the blocking effect of CE film provides more water for cement to hydrate, and hydrated cement mortar can obtain enough adhesive strength to have a firm lock with concrete substrate. HEMC can also improve the fluidity of fresh CSA cement mortars, which is also helpful for the improvement of tensile bond strength. On the other hand, modified cement mortar has favorable deformability and flexibility, which can make cement mortar commendably adaptable to the shrinking of the concrete substrate [37].

#### 3.2.5. Dry Shrinkage Rate

The dry shrinkage rate of CSA cement mortars with and without HEMCs within 28 days is illustrated in Figure 6. It can be seen that the dry shrinkage rate of CSA2 cement mortar was lower than that of CSA1 cement mortar. From Figure 6a, it can be found that all three kinds of HEMCs could increase the dry shrinkage rate of CSA1 cement mortar. With the increase in curing age, the increase of the dry shrinkage rate of CSA1 cement mortar gradually slowed down, especially after 14 days. The results show that HEMC1 does not influence the dry shrinkage rate of CSA1 cement mortar significantly. As can be seen from Figure 6b, all three kinds of HEMCs led to an increase in the dry shrinkage of CSA2 cement mortar. The dry shrinkage rate of CSA2 cement mortar and modified CSA2 cement mortar increased rapidly before 7 days. After 7 days, the increase in the dry shrinkage rate of CSA2 cement mortar and modified CSA2 cement mortar gradually slowed down, even becoming a nearly horizontal line after 14 days. At 28 days, no significant difference could be observed among the three types of HEMC-modified CSA2 cement mortars and plain CSA2 cement mortar.

During the process of mortar curing, water loss can lead to dry shrinkage [38]; the pore size distribution of the mortar also influences dry shrinkage [39]. The slight increase in the dry shrinkage rate of CSA cement mortars is probably related to the change in pore size distribution, which will be discussed in the following section, since the HEMCs have excellent water-retention capacity. The dry shrinkage rate of CSA2 cement mortar was slightly lower than that of CSA1 cement mortar. It is probably related to the content of ye’elimite. According to the references, the high content of ye’elimite in CSA cement leads to high dry shrinkage when the content of gypsum is similar [40]. The dry shrinkage rate of all the CSA cement mortars is much smaller compared with Portland cement mortar [41]. It can be ascribed to the compensation shrinkage of AFt in CSA cement [42]. Above all, it can be concluded that the influence of the type of HEMC on the dry shrinkage rate can be negligible, but the impact of cement itself is more significant.

### 3.3. Pore Size Distribution and Porosity

The pore size distributions of CSA cement mortars with and without HEMCs obtained from MIP tests are presented in Figure 7. In the pore size distribution curves of plain CSA1 cement mortar, there are two main peaks at around 100 nm and 300~400 nm, respectively, but in plain CSA2 cement mortars, only one peak appears at around 250 nm. The addition of HEMCs leads to the rapid increase in the intrusion mercury volume. Specifically, HEMCs increase the amount of the pores in CSA1 cement mortar, especially the pores in the range of 140 nm~1.25 µm. The amount of the pores with a diameter of >50 nm in CSA2 cement mortar is also increased by the additional HEMCs, especially the pores with a diameter around 700 nm. Silva et al. [43] reported that pores with a size of >50 nm affect the strength of cement mortar significantly, and obviously, HEMCs increase the amount of this kind of pores, which results in the decrease of the compressive strength and flexural strength. The pores with a size of <50 nm affect the shrinkage of cement mortar and it was found that HEMCs had a minor effect on the amount of this kind of pores, which results in the slight increase in the dry shrinkage rate of CSA cement mortars.

Porosity is always regarded as the primary element that affects the strength of cement mortar [44,45]. The porosity of CSA cement mortars with and without HEMCs at 28 days is illustrated in Figure 8. It can be seen that the porosity of plain CSA1 cement mortar was lower than that of plain CSA2 cement mortar, which is probably the reason why the strength of CSA1 cement mortar was higher. The addition of HEMCs led to a significant increase in the porosity of CSA cement mortar, which resulted in the decline of compressive strength. The MS/DS had no significant impact on the porosity of CSA cement mortar.

There was no significant difference in porosity among the six HEMC-modified cement mortars, but some strengths of the mortars were different, especially the flexural strength and tensile bond strength. It was found that the formation of a CE-bridge between Ca(OH)_2_ crystals probably had a contribution on the improvement of flexural strength of Portland cement mortar [46,47,48]. But the dosage of CE in the literature is 0.7% higher than that in this investigation. The reason why HEMCs with different MS/DS and PAAm modifications have different impacts on the flexural strength and especially tensile bond strength of CSA cement mortar needs to be further investigated.

## 4. Conclusions

CSA cement with higher ye’elimite content exhibits higher strengths, including compressive, flexural and tensile bond strength, lower porosity but a higher dry shrinkage rate.

HEMCs can improve the workability of CSA cement mortars, leading to an increase in the water-retention rate, consistency, and fluidity of fresh mortars. HEMCs decrease the compressive and flexural strength of the hardened mortar but increase the flexibility and tensile bond strength significantly and increase the dry shrinkage rate slightly. The addition of HEMCs introduces micron-level pores into CSA cement mortar and leads to an increase in porosity, but it has a minor effect on the amount of pores with a size smaller than 50 nm.

The water-retention capacity of HEMC in CSA cement mortars does not depend on the MS/DS or PAAm modification, and the MS/DS and PAAm modification have a minor effect on the effect of HEMC on the flexibility, dry shrinkage, and porosity of CSA cement mortars. However, HEMCs with different MS/DS do have different impacts on compressive/flexural strength, and the impacts are related to the type of CSA cement used. PAAm cannot change the effect of HEMC on compressive strength, but it can change the effect of HEMC on flexural strength. The MS/DS and PAAm modifications have a significant effect on the tensile bond strength.

Finding out the function mechanism of different HEMCs in different CSA cement mortars is an ongoing work. However, HEMCs have adverse impacts on compressive and flexural strength that deserves consideration and to find ways to solve when HEMCs are applied in commercial mortar.

## Figures and Tables

**Figure 1 molecules-26-02136-f001:**
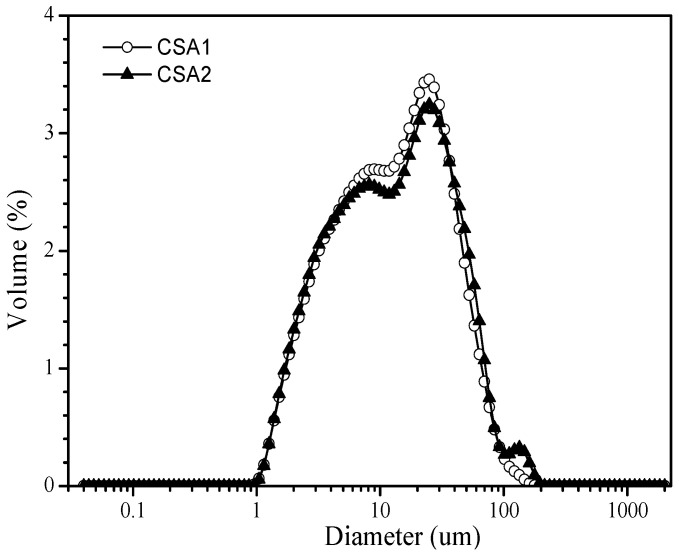
Particle size distribution of cements.

**Figure 2 molecules-26-02136-f002:**
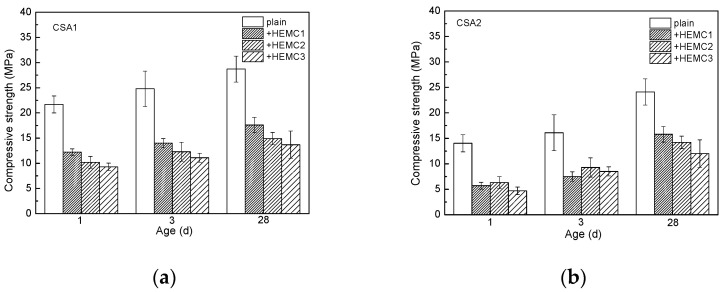
Compressive strength of CSA cement mortars with and without HEMCs at different curing ages: (**a**) CSA1, (**b**) CSA2.

**Figure 3 molecules-26-02136-f003:**
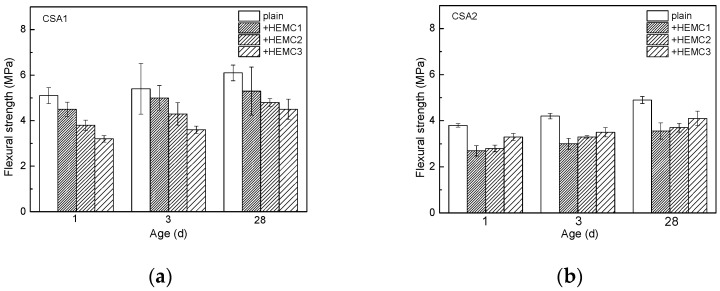
Flexural strength of different CSA cement mortars with and without HEMCs at different curing ages: (**a**) CSA1, (**b**) CSA2.

**Figure 4 molecules-26-02136-f004:**
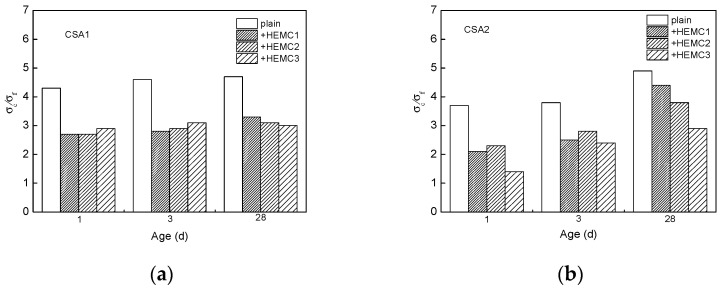
The ratio of compressive strength to flexural strength of CSA cement mortars with and without HEMCs at different curing ages: (**a**) CSA1, (**b**) CSA2.

**Figure 5 molecules-26-02136-f005:**
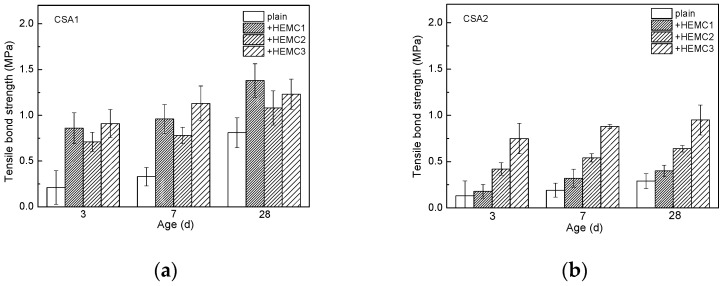
Tensile bond strength of CSA cement mortars with and without HEMCs at different curing ages: (**a**) CSA1, (**b**) CSA2.

**Figure 6 molecules-26-02136-f006:**
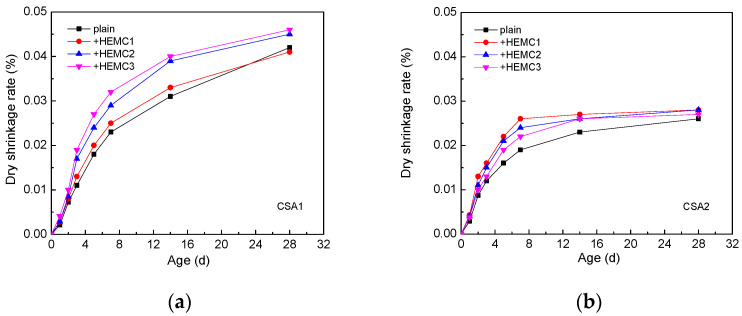
The dry shrinkage rate of CSA cement mortars with and without HEMCs within 28 days: (**a**) CSA1, (**b**) CSA2.

**Figure 7 molecules-26-02136-f007:**
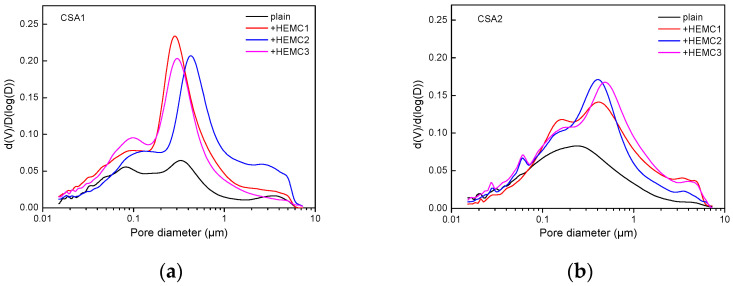
The pore size distribution of CSA cement mortars with and without HEMCs at 28 days: (**a**) CSA1, (**b**) CSA2.

**Figure 8 molecules-26-02136-f008:**
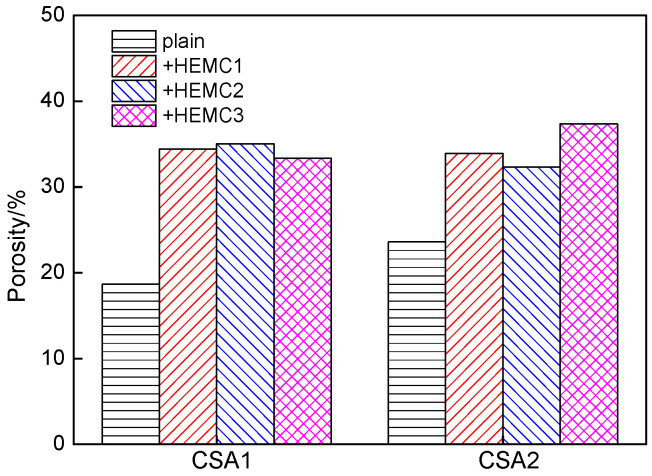
The porosity of CSA cement mortars with and without HEMCs at 28 days.

**Table 1 molecules-26-02136-t001:** Chemical composition of cements Wt/%.

	SiO_2_	Al_2_O_3_	Fe_2_O_3_	CaO	MgO	SO_3_	Na_2_O	K_2_O	TiO_2_	P_2_O_5_
CSA1	13.00	17.80	1.90	44.90	2.66	12.60	0.11	0.41	0.94	0.16
CSA2	18.00	13.60	2.52	44.50	3.58	13.10	0.25	0.42	0.77	0.28

**Table 2 molecules-26-02136-t002:** Mineral composition of cements Wt/%.

	C_4_A_3_$\bar{S}$	C_2_S	C_4_AF	C_12_A_7_	CT	Gypsum	Anhydrite	Amorphous
CSA1	28.4	17.9	4.0	1.5	2.6	1.5	9.6	19.4
CSA2	15.7	25.1	3.5	3.6	1.3	1.9	12.4	30.0

**Table 3 molecules-26-02136-t003:** Physico-chemical properties of HEMC1 and HEMC2.

	HEMC1	HEMC2
Viscosity, 2% solution, 20 °C, Haake RotoVisco (mPa·s)	35,500	39,100
Average fiber width (µm)	7	18
Average fiber length (µm)	290	126
Molecular mass, numbers medium (g·mol^−1^)	373,000	577,400
Molecular mass, weight medium (g·mol^−1^)	742,500	759,500
Drying loss (Wt %)	3.9	5.6
Sodium chloride (Wt %)	2.37	2.81
DS	1.41	1.57
Total MS/DS	1.6	1.90

Viscosity, molecular mass, sodium chloride, DS, and total MS/DS measured by DOW Chemical, Germany, Bomlitz.

**Table 4 molecules-26-02136-t004:** Water-retention rate, consistency and fluidity of fresh CSA cement mortars with and without HEMCs.

	Water Retention Rate/%	Consistency/mm	Fluidity/mm
CSA1	87.0	20	130
CSA1 + HEMC1	99.8	38	136
CSA1 + HEMC2	99.7	40	142
CSA1 + HEMC3	99.9	35	144
CSA2	86.2	24	122
CSA2 + HEMC1	99.4	46	147
CSA2 + HEMC2	99.7	34	148
CSA2 + HEMC3	99.6	40	147

## Data Availability

Date sharing not applicable.

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
