# Peer review of "Properties of Calcium Sulfoaluminate Cement Mortar Modified by Hydroxyethyl Methyl Celluloses with Different Degrees of Substitution"

_molecules, 2021, doi:10.3390/molecules26082136_

Round 1
Reviewer 1 Report
In the paper titled "Properties of calcium sulfoaluminate cement mortar modified by hydroxyethyl methyl celluloses with different degrees of substitution", the authors compared the influence of high and low MS/DS and PAAm modified hydroxy-ethyl methyl cellulose (HEMC) on the properties of CSA cement mortar with different content of ye’elimite. They also analyzed the porosity and pore size distribution of different CSA cement mortars by MIP. The study is interesting. Here are some suggestions for this article.
- Page 2, Line 81, why do you choose CSA1 and CSA2 cements? Please give your reasons.
- Page 4, Line 112, what is the meaning of “m1, m2, m3, m4 and α” in equation (1)?
The same questions appear in equation (2) and equation (3).
- The literature review on “pore size distribution and porosity” is incomplete, leading to a lack of depth in the analysis in the discussion section of the following text.
- In “3. Results and discussion”, regarding the changes law of “Hardened state properties”, such as compressive strength, flexural strength and ratio of compressive strength to flexural strength, dry shrinkage rate, etc., please give an in-depth analysis rather than a simple description.
- The conclusion needs to be further refined.
Reviewer 2 Report
The article "Properties of calcium sulfoaluminate cement mortar modified by hydroxyethyl methyl celluloses with different degrees of substitution" is well written and it presents interesting results. However, before publication, it is recommended to include the following aspects:
- Include criteria used for selection of the mix design evaluated. “The mortar specimens were prepared with HEMC/cement mass ratio (mHEMC/mc) of 0.3%, water/cement mass ratio of 0.60 and cement/sand mass ratio of 3:7.”
- Include further research in the conclusions. For example, a topic such as the decrease in compressive and flexural strengths seems to be improved and studied.
- Check the use of capital letters for Table 4 (line 168) and Table 2 (197).
